# Genome-Wide Analysis of the Caffeoyl Coenzyme A-O-Methyltransferase (*CCoAOMT*) Gene Family in *Platycodon grandiflorus* (Jacq.) A. DC. and the Potential Regulatory Mechanism in Response to Copper Stress

**DOI:** 10.3390/ijms26104709

**Published:** 2025-05-14

**Authors:** Junbai Ma, Shan Jiang, Lingyang Kong, Lengleng Ma, Xinxin Wang, Meitong Pan, Chenliang Li, Shumin Huang, Xiubo Liu, Wei Ma, Weichao Ren

**Affiliations:** 1College of Pharmacy, Heilongjiang University of Chinese Medicine, Harbin 150040, China; 15114516116@163.com (J.M.); 17390928032@163.com (S.J.); hljkly970219@163.com (L.K.); 18739466784@163.com (L.M.); a1302196384@126.com (X.W.); meitong_pan@163.com (M.P.); l7174080625@163.com (C.L.); hsm182375@163.com (S.H.); 2College of Jiamusi, Heilongjiang University of Chinese Medicine, Jiamusi 154007, China; zyylxb@foxmail.com

**Keywords:** *Platycodon grandiflorus* (Jacq.) A. DC., copper stress, CCoAOMT gene family, lignin

## Abstract

In recent years, copper pollution has gradually become one of the major problems of soil environmental pollution. Lignin plays an important role in plant resistance to biotic and abiotic stresses. CCoAOMT is a key enzyme in the lignin biosynthesis process. In this study, the *CCoAOMT* gene family members of *Platycodon grandiflorus* were identified by bioinformatics methods, and their basic characteristics and potential functions were analyzed. The results showed that five members of the *PgCCoAOMT* gene family were identified in *P. grandiflorus*, with protein lengths ranging from 246 to 635 amino acids, and were evenly distributed on four chromosomes. Phylogenetic analysis indicated that the *PgCCoAOMT* gene family was divided into two subclades, namely Clade1a, Clade1b, Clade1c, Clade1d, and Clade2. The cis-regulatory element analysis of the promoter revealed that the *PgCCoAOMT* members contained a large number of cis-regulatory elements responsive to stress, and conjecture *PgCCoAOMT2*, *PgCCoAOMT4*, and *PgCCoAOMT5* were involved in the lignin synthesis. The qRT-PCR results showed that, within 5 days of copper stress treatment, except for the *PgCCoAOMT4* gene, the other genes exhibited different expression levels. Furthermore, the expression levels of all five *PgCCoAOMT* genes increased significantly at 7 days of treatment. With the increase in the number of days of treatment, the content of lignin in the seedings of *P. grandiflorus* showed a trend of increasing first and then decreasing under copper stress. In general, in the copper stress treatment of 1–3 days, the transcriptional inhibition of *PgCCoAOMT1* and *PgCCoAOMT3* and the increase in lignin content contradicted each other, suggesting that there was post-translational activation or alternative metabolic pathways compensation. Meanwhile, in the 7-day treatment, the coordinated up-regulation of the genes was accompanied by the failure of lignin synthesis, which pointed to the core bottleneck of metabolic precursors depletion and enzyme activity inactivation caused by root damage. Research objective: This study reveals the expression level of the *PgCCoAOMT* gene in the seedings of *P. grandiflorus* under copper stress, providing a theoretical basis for elucidating the mechanism of *P. grandiflorus* response to copper stress and for subsequent improvement of root resistance in *P. grandiflorus*.

## 1. Introduction

In the context of extensive human activities and rapid economic development, heavy-metal soil pollution has increasingly emerged as a global issue, posing a significant challenge to the survival of animals, plants, and humans [1]. Simultaneously, heavy-metal pollution has gradually become one of the most prominent concerns in the quality control of traditional Chinese medicine [2,3,4]. For medicinal plants, heavy metals in the soil have adverse impacts on plant growth and metabolism, severely affecting the quality and safety of Chinese medicinal materials [5]. Currently, heavy-metal pollution involving cadmium, lead, and mercury has received considerable attention from scholars at home and abroad. Moreover, soil copper pollution has gradually become one of the primary issues in soil environmental pollution [6]. Copper is an essential trace element for plant growth and development and is extensively involved in various physiological and biochemical processes of plants, including chlorophyll formation, photosynthesis, protection against oxidative stress, and protein and carbohydrate metabolism [7]. When the copper content in plants is insufficient, the plant leaf surface begins to turn black and green and may eventually lead to plant death. In contrast, once the concentration of copper ions exceeds the critical threshold, it will exhibit toxic effects on plants. Ion homeostasis in plants will be interrupted, and in severe cases, it may even cause plant death [8,9,10].

*Platycodon grandiflorus* (Jacq.) A. DC., a perennial herbaceous plant within the genus Platycodon of the *Campanulaceae* family, has a wide distribution in the northeast, north, east, central, and southwest regions of China [11]. In Heilongjiang Province, it is ranked among the first batch of genuine regional medicinal materials, with its dried root being the medicinally used part. Its application in traditional medicine dates back to ancient times. It was first documented in the Shennong Bencao Jing. Subsequently, it has been continuously described in numerous medical books and classics throughout different historical dynasties. Modern chemical and pharmacological studies have revealed that *P*. *grandiflorus* is abundant in diverse bioactive components, including saponins, flavonoids, phenols, polyacetylenes, and polysaccharides. These constituents play a pivotal role in anticancer, anti-inflammatory, antioxidant, anti-diabetes, and liver protection functions [12,13,14,15,16,17]. In addition to its remarkable medicinal value, *P. grandiflorus* also holds a significant position in the food domain. It is one of the traditional pickled vegetables in North Korea, South Korea, Japan, and northeast China [18]. In 2002, it was inscribed in the first batch of the homologous list of medicine and food in China. Furthermore, in 2021, it was designated as a national geographic indication of agricultural products in the vegetable classification. As a result, the market demand for *P. grandiflorus* has been on a steady upward trajectory year by year. However, this plant demonstrates a strong capacity for soil copper enrichment. This phenomenon has become a dominant problem in the production of traditional Chinese medicine, severely affecting both the yield and quality of *P. grandifloras* [19]. To date, knowledge about the molecular mechanisms through which medicinal plants respond to copper stress remains insufficient. Thus, exploring the molecular mechanism underlying the copper tolerance of *P. grandiflorus* is of utmost importance. Such an exploration will provide a crucial theoretical basis for the breeding of superior-quality varieties with enhanced copper tolerance capabilities.

Caffeoyl coenzyme A-O-methyltransferase (CCoAOMT) belongs to the S-adenosine methionine (SAM) family and is a type of methyl transferase. As an SAM-dependent O-methyltransferase, this enzyme can transfer the methyl group from the SAM to the benzene ring. It can catalyze a variety of compounds, including phenylpropanoids, flavonoids, anthocyanins, and coumarins [20,21,22,23]. CCoAOMT serves as a pivotal methyltransferase in the lignin biosynthesis pathway in plants. It participates predominantly in lignin biosynthesis through the phenylpropanoid pathway [24]. In particular, CCoAOMT plays a fundamental role in lignin biosynthesis by facilitating the conversion of caffeoyl-CoA (CCoA) to feruloyl-CoAF (CoA) [25]. The *CCoAOMT* gene was first proposed in *Petroselinum crispum* as a possible caffeoyl coenzyme A-O-methyltransferase, and then it was first confirmed that it was involved in lignin biosynthesis in Zinnia [26,27]. The *CCoAOMT* gene family has been identified and analyzed comprehensively in numerous plant species, such as *Arabidopsis thaliana*, *Oryza sativa*, *Nicotiana tabacum*, and *Gossypium hirsutum* [28,29,30]. However, to date, there have been no reports on the *CCoAOMT* gene in *P. grandiflorus*. Studies have shown that lignin plays an important role in the formation of plant cell walls and enhances the ability of plants to resist biotic and abiotic stresses [31]. The *CCoAOMT1* gene in *A*. *thaliana* regulates the accumulation of H_2_O_2_ and the responses of ABA and ROD signaling pathways to drought stress by modulating the drought resistance [32]. In maize, *ZmCCoAOMT2* regulates the content of H-lignin and programmed cell death (PCD), playing a crucial role in resistance against certain diseases [33]. Therefore, investigating the *PgCCoAOMT* gene could provide valuable information on the mechanism of the lignin synthesis pathway in *P. grandiflorus.* This exploration may also have the promise of harnessing this gene to enable *P. grandiflorus* to better withstand copper stress.

## 2. Results

### 2.1. Identification and Analysis of Members of the PgCCoAOMT Gene Family

Five predicted *PgCCoAOMT* genes were identified from the *P. grandiflorus* genome, and all PgCCoAOMTs have the conserved O-methyltransferase (PF01596) domain. In this study, amino acid length, theoretical isoelectric point, molecular weight, and subcellular location of PgCCoAOMT were analyzed (Table 1). The results showed that the longest PgCCoAOMT protein (PgCCoAOMT4) contained 635 amino acid residues, the shortest PgCCoAOMT protein (PgCCoAOMT1) contained 246 amino acid residues, and the remaining three sequences had about 300 amino acid residues. The molecular weight of PgCCoAOMT4 was 69.76 k Da, and other proteins were within 40 k Da. The theoretical isoelectric point is between 5.56 (PgCCoAOMT1) and 9.15 (PgCCoAOMT3). Four PgCCoAOMT proteins were located in the cytoplasm and one in the chloroplast.

### 2.2. Chromosome Localization Analysis, Multiple Sequence Alignment, and Phylogenetic Analysis of the PgCCoAOMT Gene

Four genes in the *PgCCoAOMT* gene family were evenly distributed on four chromosomes of *Platycodon grandiflorum*, and one gene (*PgCCoAOMT5*) was distributed on contig00572 (Figure 1a). There is one *CCoAOMT* gene on chromosomes chr2, chr3, chr8, and chr9. They are named *PgCCoAOMT1* to *PgCCoAOMT5*, depending on their distribution on the chromosome.

Multiple sequence alignment of PgCCoAOMT protein sequences was performed to detect the conserved domain, and it was found that PgCCoAOMT had all of the characteristic elements of the *CCoAOMT* genes and the sequences were highly conserved. According to the results of Joshi and Chiang (1998), the conserved domain was named motif A-H [34]. The motif A-C is the methyltransfer_3 domain, which is a common component of plant methylation. Motif D-H is the SAM binding sites, but it was found that the deletion or mutation of the base of the gene conserved sequence component is shown in Figure 1b [35].

In this study, the five identified *PgCCoAOMT* members were used to construct a phylogenetic tree together with 58 *CCoAOMT* members from *A. thaliana*, *B. nivea*, *C. capsularis*, *C. sinensis*, *G. max*, *H. cannabinus*, *I. indigotica*, *O. sativa*, and *T. cacao*. In the phylogenetic tree, the members of *PgCCoAOMT* are divided into two subfamilies, Clade1 and Clade2, according to the classification method of the *CCoAOMT* gene family of *A. thaliana*. Among them, Clade1 can be further divided into Clade1a, Clade1b, Clade1c, and Clade1d, and have 30, 19, four, and eight members, respectively. *PgCCoAOMT1* and *PgCCoAOMT4* belong to Clade1a, *PgCCoAOMT2* and *PgCCoAOMT5* belong to Clade1b, and *PgCCoAOMT3* belongs to Clade1d. No specific populations were found in the phylogenetic trees of the *PgCCoAOMT* gene (Figure 1c).

### 2.3. PgCCoAOMT Gene Structure, Conserved Motifs, and Intron–Exon Analysis

As can be seen in Figure 2, the *PgCCoAOMT1*, *PgCCoAOMT2*, and *PgCCoAOMT5* genes contain only four introns, and it is found that the gene structure of a subfamily gene member (*PgCCoAOMT2* and *PgCCoAOMT5*) is relatively similar, suggesting that these two genes have similar functions.

The structure of each gene, especially the number and distribution of exons and introns, may be closely related to evolution [36]. The *PgCCoAOMT3* gene structure contains eight introns; the *PgCCoAOMT4* gene structure contained the largest number of introns (11), and the difference was obvious. This indicates that the *PgCCoAOMT1*, *PgCCoAOMT2*, and *PgCCoAOMT5* genes are the most primitive *CCoAOMT* genes in *P. grandiflorus*. During the phylogenetic process of *P. grandiflorus*, *PgCCoAOMT3* and *PgCCoAOMT4* genes have undergone multiple gene splicing or insertions of fragments of genes. These two genes are gradually differentiated to adapt to new environmental changes and perform new protein functions.

Analysis of the functions of gene domains is fundamental to comprehending the regulatory mechanisms of life. Therefore, this study predicted the conserved domains and modifications of the *PgCCoAOMT* gene and found that there were some differences in the conserved domains of different members of the *CCoAOMT* gene family. For example, *PgCCoAOMT1*, *PgCCoAOMT2*, *PgCCoAOMT3*, and *PgCCoAOMT4* all contain the conserved domain PLN02589, while *PgCCoAOMT5* contains the AdoMet MTases superfamily. These results indicated that the biological function of the *PgCCoAOMT5* gene may be significantly different from that of *PgCCoAOMT1*, *PgCCoAOMT2*, *PgCCoAOMT3*, and *PgCCoAOMT4*. In addition, we analyzed five conserved motifs of the PgCCoAOMT protein sequence using the MEME online tool. It should be noted that all PgCCoAOMT proteins have Motif1, Motif2, Motif3, and Motif4, and PgCCoAOMT4 contains two groups of Motif1–4, indicating that Motif1–4 is highly conserved in the *CCoAOMT* gene family. Motif1–4 may play an important role in the function and evolution of the *CCoAOMT* gene. Moreover, the number and sequence similarity of conserved motifs of the *CCoAOMT* gene located in the same subfamily are high, also indicating that members of the *CCoAOMT* gene family located in the same subfamily have similar biological functions. PgCCoAOMT2 and PgCCoAOMT5 contain Motif5 and are presumed to have similar functions.

### 2.4. Analysis of Cis-Regulatory Element of the PgCCoAOMT Gene Promoter

The results of the promotion prediction showed that a total of 50 cis-regulatory elements were detected in the promoters of the *PgCCoAOMT* gene, and all promoter regions of the *PgCCoAOMT* gene contained four types of cis-regulatory elements: basic promoter elements, elements related to biotic and abiotic stress, elements related to hormone response, and elements related to plant growth and development. In this study, 29 representative cis-regulatory elements were selected for mapping and manual removal of cis-regulatory elements that could not be accurately classified. As can be seen in Figure 3, in terms of biological and abiotic stress-related elements, all *PgCCoAOMT* genes were found to have MYB binding sites and MYC binding sites, among which the *PgCCoAOMT3* gene had the most MYB binding sites (seven) and the *PgCCoAOMT5* gene had the most MYC binding sites (five). It is speculated that MYB and MYC transcription factors may directly activate the transcription of the *PgCCoAOMT* gene promoter to improve the tolerance of platycodon grandidon to biotic and abiotic stresses. All *PgCCoAOMT* genes have the drought response element (MBS) and the heat shock protein-related element (STRE). *PgCCoAOMT3* and *PgCCoAOMT4* contain anaerobic induction elements (AREs) and low-temperature response elements (LTR). *PgCCoAOMT3* has drought and high salt stress response elements (DRE core). *PgCCoAOMT1*, *PgCCoAOMT2*, and *PgCCoAOMT5* all have a defense and stress response element (TC-rich repeats). Except for the *PgCCoAOMT1* gene, all the other genes had the WUN-motif. Except for the *PgCCoAOMT4* gene, all other genes possess W-box cis-regulatory element. WRKY transcription factors are speculated to bind to the W-box cis-regulatory element in the promoter region of these four *PgCCoAOMT* genes to activate or inhibit downstream gene expression. In terms of hormone-responsive elements, all *PgCCoAOMT* genes possess abscisic acid-responsive elements (AAGAA-motif and ABREs) and ethylene-responsive elements (EREs). Except for the *PgCCoAOMT3* gene, all other genes had a salicylic acid-responsive cis-regulatory element (as-1). The *PgCCoAOMT1* gene has the GARE-motif, the *PgCCoAOMT2* gene contains the TATC-box, and the *PgCCoAOMT3* gene has the P-box. Both the GARE-motif and TATC-box are gibberellin-response elements. Both *PgCCoAOMT3* and *PgCCoAOMT4* have salicylic acid response elements (TCA-element). *PgCCoAOMT1*, *PgCCoAOMT2*, and *PgCCoAOMT4* all have MeJA methyl jasmonate response elements (TGACG-motif). In terms of plant growth and development-related elements, all *PgCCoAOMT* genes contain light response elements such as light responsiveness (Box 4, G-Box, GA-motif, GT1-motif, I-box, MRE, and Sp1). *PgCCoAOMT1*, *PgCCoAOMT2*, and *PgCCoAOMT5* have a large number of AT~TATA-box cis-regulatory elements, with 31, 20, and 24, respectively, accounting for the majority of plant growth and development-related elements. *PgCCoAOMT2* and *PgCCoAOMT5* have AT-rich element cis-regulatory elements. *PgCCoAOMT2*, *PgCCoAOMT3*, and *PgCCoAOMT5* have meristem expression regulatory elements (CAT-box), among which *PgCCoAOMT3* has the largest number of such elements.

### 2.5. Prediction of Secondary Structure of the PgCCoAOMT Protein

Proteins’ secondary structure is the fundamental basis for maintaining the structural stability and realizing the biological functions of proteins. The prediction results of the secondary structure of the *P. grandiflorus* CCoAOMT protein indicate that in the PgCCoAOMT family, the proportions of secondary structures are as follows: alpha helix ranges from 33.57% to 39.37%, extended strand from 12.28% to 17.89%, beta turn accounts for 0%, and random coil ranges spans 43.50% to 52.14% (Table 2). The protein sequences of PgCCoAOMT2 and PgCCoAOMT5 are very similar, but the results show that there are certain differences in the secondary structure of the two proteins, suggesting that PgCCoAOMT2 and PgCCoAOMT5 may have different functions.

### 2.6. Analysis of CCoAOMT Gene Expression in P. grandiflorus Under Copper Ion Stress

The expression of the *PgCCoAOMT* gene under copper stress at 1 d, 3 d, 5 d, and 7 d was verified by qRT-PCR analysis compared with the CK group (Figure 4). The results showed that all genes except *PgCCoAOMT4* showed different expression patterns after 5 days of copper stress treatment. The expression of *PgCCoAOMT1* and *PgCCoAOMT3* shows a trend of inhibition followed by activation, suggesting that they are involved in the staged repair process. The early down-regulation of these genes under copper stress treatment may be for resource conservation in metabolism, while the later up-regulation is to cope with the cell wall damage caused by copper ion permeation. Both *PgCCoAOMT2* and *PgCCoAOMT5* showed down-regulated expression. It is speculated that the promoter regions of these two genes may contain stress-sensitive elements. In the early stage of copper stress, they are inhibited by transcription factors and prioritize the allocation of resources to the antioxidant pathway. The expression levels of five *PgCCoAOMT* genes increased significantly after 7 days of treatment. It is speculated that the metabolic collapse caused by root damage resulting from long-term stress leads to the failure of gene expression up-regulation.

### 2.7. PgCCoAOMT Protein Interaction Network Prediction

The results showed (Figure 5) that except for the PgCCoAOMT3 protein, which did not predict the relevant interaction nodes, there were about eight proteins directly interacting with other *PgCCoAOMT* gene family members, most of which were involved in the lignin biosynthesis pathway, and there were 22 interaction relationships. It is speculated that PgCCoAOMT1, PgCCoAOMT2, PgCCoAOMT4, and PgCCoAOMT5 proteins are widely involved in lignin biosynthesis.

### 2.8. The Trend of Lignin Content Changes in the Tissue Samples of P. grandiflorus

Based on the sample weight, the lignin content showed a trend of increasing first and then decreasing. Among them, the lignin content reached the peak at 1 day of stress, demonstrating the characteristic of rapid stress response. The lignin content began to decline after 1 day and was particularly lower than that of the control group at 5 days and 7 days, indicating that the root damage was severe in the later stage.

## 3. Discussion

Lignin is the primary component of cell wall skeleton and plays a key role in plant responses to biotic and abiotic stresses [37]. CCoAOMT is a key enzyme in the lignin biosynthesis pathway of plants. It has been extensively studied in *A. thaliana*, *O. sativa*, populus, cotton, and many other plants, but it has not been reported in *P. grandifloras* [29,30,35,38]. Therefore, in this study, five *PgCCoAOMT* genes were selected based on genomic data from *P. grandiflorus*, and bioinformatic analysis was performed on them, including physicochemical property analysis, gene structure, conserved motif analysis, intron–exon analysis, phylogenetic analysis, promoter cis-regulatory elements analysis, etc., providing a theoretical basis for further research on the potential functions of CCoAOMT.

### 3.1. Effect of Copper Treatment on Appearance and Morphology of P. grandiflorus Seedlings and the Changing Trend of Lignin Content

This study focused on the seedlings of *P. grandiflorus* and analyzed the trend of lignin content changes under continuous copper stress (200 mM CuSO_4_·5H_2_O) for 7 days. Meanwhile, the mechanism of *P. grandiflorus* seedlings resisting copper stress was preliminarily revealed. In Figure 6, we revealed the changes in lignin content, showing that the lignin content of *P. grandiflorus* seedlings reached the peak on the first day of copper stress treatment, slightly decreased on the third day, but still higher than the CK group, and there was no obvious difference in the appearance of the seedlings. It was speculated that plants may accumulate lignin to strengthen cell walls and restrict ion permeability. However, on the fifth and seventh days of copper stress treatment, the lignin content showed a significant downward trend, significantly lower than the CK group, and reached the lowest level on the seventh day. The morphology of the seedlings changed significantly, with the leaf tips and margins gradually turning yellow and then withering, and even some parts died and fell off; the roots showed symptoms of root rot, with the number of adventitious roots continuously decreasing, the root system gradually rotting, and even the rot spreading to the main root (Figure 6).

### 3.2. Bioinformatics Analysis of PgCCoAOMT Gene Family

The physical and chemical properties of the *CCoAOMT* gene family showed that there were differences in the number of sequence amino acids in the range of 246~635 aa and molecular weight in the range of 27.66~69.76 kD. Subcellular localization prediction results showed that the CCoAOMT protein in *P. grandiflorus* was mainly localized to the cytoplasm and chloroplasts, which was the same as the predicted results in Cotton and Jute [30,39]. The subcellular localization of the PgCCoAOMT3 protein is predicted in chloroplasts, which is the subcellular structure with the highest copper content in plants. At the same time, copper is an important structural component of plastocyanin and chlorophyll, so copper plays a crucial role in the smooth process of the photosynthetic electron transport chain and photosynthesis. According to qRT-PCR analysis results, the expression level of *PgCCoAOMT3* was higher than that of the other four genes after 7 days of copper stress treatment. It could be speculated that copper stress affects photosynthesis by influencing chlorophyll synthesis. Therefore, the expression of the *PgCCoAOMT3* gene changed significantly.

According to phylogenetic analysis, the *CCoAOMT* gene family of *P. grandiflorus* was divided into two subgroups, Clade1a, Clade1b, Clade1c, Clade1d, and Clade2, according to the classification method of the *A. thaliana* and *O. sativa* model plants, and all members had the SAM domain (PF01596). It belongs to the AdoMet-MTases superfamily. Figure 1c shows that *P. grandiflorus* is the only Platycodon species in the Platycodon family [11]. It is more distantly related to the nine selected species but does not form new clades like *O. sativa*, and it is speculated that PgCCoAOMT has some different functional properties. Differences in gene structure and conserved motifs can lead to different CCoAOMT proteins, and these characteristics may be closely related to the adaptation of *P. grandiflorus* to abiotic stress such as copper stress and the formation of herbal quality. Each protein sequence contains a different number and type of conserved motif, which can reveal a different function of each gene. Different subfamilies have different gene structures or conserved motifs. For example, *PgCCoAOMT2* and *PgCCoAOMT5* have conserved Motif5 that other subfamily members do not have and have similar gene structures and conserved domains to other subfamily members. It is speculated that other subfamilies may have lost conserved motifs during the evolutionary process. However, it retained other features of the *CCoAOMT* gene family.

Transcriptional regulation plays a dominant role in the control of gene expression. Gene expression is primarily governed by cis-regulatory elements located in the gene promoter region, and these cis-regulatory elements within the upstream regulatory sequences can modulate responses to stress and hormones, as well as influence growth and development. The analysis of cis-regulatory elements of the *PgCCoAOMT* gene promoter showed that all genes contained photoresponsive elements. In addition, a large number of stress-related elements and hormone-responsive elements were identified, suggesting that the *PgCCoAOMT* gene may play a key role in growth and development, hormone response, and stress response.

### 3.3. The Potential Role of PgCCoAOMT Gene Under Copper Stress

The qRT-PCR results indicated (Figure 4) that within 5 days of copper stress treatment, members of the *PgCCoAOMT* gene family showed different expression patterns. The expression of *PgCCoAOMT1* and *PgCCoAOMT3* showed a trend of inhibition followed by activation, while *PgCCoAOMT2* and *PgCCoAOMT5* showed continuous down-regulation, and the expression of *PgCCoAOMT4* showed no significant change. Notably, at the long-term stress (7 days) stage, the expression levels of all *PgCCoAOMT* genes were significantly up-regulated. However, the lignin content showed a trend of increasing first and then decreasing. This lag in gene expression and the rapid accumulation of lignin content jointly suggested that there was a multi-level regulatory network in the seedings of *P. grandiflorus* under copper stress. Therefore, we speculate that under 1–3 days of stress treatment, plants initiated a rapid response, showing rapid accumulation of lignin content. This might be dependent on the early accumulation of precursor substances or the spontaneous activation of other biosynthetic pathways of lignin. The brief inhibition of *PgCCoAOMT1* and *PgCCoAOMT3* might be a resource optimization strategy, allocating metabolic fluxes preferentially to the antioxidant system to resist ROS bursts. At 3–5 days of stress treatment, metabolic imbalance occurred in the seedings of *P. grandiflorus*. *PgCCoAOMT2* and *PgCCoAOMT5* showed continuous inhibition, which might lead to the blockage of the key step of lignin methylation and affect the subsequent precursor supply, thereby causing the gradual decrease in lignin content. Finally, under 5–7 days of copper stress treatment, the metabolic network in the plant body showed a systemic collapse. Although the overall expression of *PgCCoAOMT* genes was significantly up-regulated, it was unable to reverse the lignin synthesis and accumulation in the plant body. Specifically, the lignin content was significantly lower than the CK group on day 7, and leaf scorch and root rot diseases occurred.

## 4. Materials and Methods

### 4.1. Experimental Material

The plant materials of this experiment were mature seeds of *P. grandiflorus* collected from the Botanical Garden of Heilongjiang University of Chinese Medicine on 12 October 2023. *P. grandiflorus* genome data can be downloaded online (https://theragenetex.com) [40]. The protein sequence information of *A. thaliana* CCoAOMT was obtained from the TAIR database [41].

### 4.2. Hydroponics Test and Copper Stress Treatment of P. grandiflorus

Large and full *P. grandiflorus* seeds were sown in the soil: vermiculite = 3:1 seedling pot, and they were cultivated in the greenhouse of Heilongjiang University of Chinese Medicine until the seedlings grew to 5 true leaves. Seedlings with good growth status were selected and transplanted into incubators containing 1/2 Hoagland nutrient solution for reducing transplant shock, and 1 plant per hole was planted. The stems of the plants were wrapped with a sponge and the nutrient solution was changed every 2 days. After 7 days of acclimatization of transplanted seedlings, the copper stress treatment test was carried out. Copper was administered in the form of 200 mM CuSO_4_·5H_2_O, and the same volume of nutrient solution was used as a blank control. The samples were collected at 1 d, 3 d, 5 d, and 7 d after treatment, and liquid nitrogen was frozen in an ultralow-temperature refrigerator at −80 °C. The preserved materials would be used for subsequent analysis of the expression patterns of target genes under stress.

### 4.3. Identification of Members of the CCoAOMT Gene Family, Analysis of Physical and Chemical Properties, Subcellular Localization, and Chromosomal Localization of P. grandiflorus

Using *A. thaliana* protein sequences as bait sequences, snake bed protein sequences were searched by bidirectional blast comparison using TBtools v2.210 software, and preliminary candidate sequences of members of the *PgCCoAOMT* gene family were obtained [42]. The sequences obtained were integrated by two methods to identify the final members of the *P. grandiflorus CCoAOMT* gene family. On the one hand, the conservative InterPro database (https://www.ebi.ac.uk/interpro/search/sequence/, accessed on 5 November 2024) was used to download the *CCoAOMT* gene structure domain hidden Markov model (PF01596). Then, the candidate protein sequences of the *CCoAOMT* gene family were screened by HMMER3.1 software (e-value < 1 × 10^−20^) [43,44]. On the other hand, the NCBI CDD online site was used to manually eliminate redundant sequences [45]. The sequences obtained by the above two methods were integrated to obtain the final members of the *CCoAOMT* gene family of *P. grandiflorus*. The online Expasy tool was used to analyze the molecular weight (MW) and theoretical isoelectric point (PI) of the predicted PgCCoAOMT protein [46]. Subcellular localization prediction of the *PgCCoAOMT* gene obtained by CELLO v.2.5 was performed [47]. By using the gff3 annotation file of the *P. grandiflorus* genome, the location information of required genes was extracted, and the Gene Density Profile and Gene Location Visualize from GTF/GFF functions of TBtools software were used. The *P. grandiflorus CCoAOMT* gene family was located on the corresponding chromosome.

### 4.4. Multiple Sequence Alignment of CCoAOMT Protein in P. grandiflorus and Phylogenetic Analysis of CCoAOMT Gene Family in 10 Different Plants

To determine whether the *P. grandiflorus CCoAOMT* gene family had all the characteristic elements of the *CCoAOMT* genes, 5 PgCCoAOMT protein sequences were compared using DNAMAN 7 software with default parameters.

The 58 protein sequences of CCoAOMT from *A. thaliana* (7), *Boehmeria nivea* (4), *Corchorus capsularis* (8), *Camellia sinensis* (10), *Glycine max* (7), *Hibiscus cannabinus* (1), *Isatis indigotica* (8), *O. sativa* (6), and *Theobroma cacao* (7) were aligned with the identified CCoAOMT protein sequence of *P. grandiflorus* using the MUSCLE program for multiple sequence alignment. Using MEGA-11.0 software to build Neighbor Joining (NJ), set the bootstrap value to 1000 repeats, and export the Newick Tree file [48]. According to the classification method of the AtCCoAOMT gene family, all the identified CCoAOMT members of *P. grandiflorus* were classified into subfamilies. Visualization and beautification of the evolution tree of the system are completed using the Evolview v3 online software [49].

### 4.5. Analysis of CCoAOMT Gene Structure, Conserved Motif, and Intron–Exon in P. grandiflorus

The structural information of the *PgCCoAOMT* gene was extracted from the GFF annotation file of the *P. grandiflorus* genome. The authors analyzed the conserved motif of the *P. grandiflorus CCoAOMT* gene using the MEME online website. The number of motifs was set to 5 and the other parameters were set to default values. The result file in mast.xml format was downloaded [50]. The conserved domain of the PgCCoAOMT protein was predicted using the NCBI CDD database [45]. Finally, combined with the phylogenetic tree of PgCCoAOMT and the gene annotation file (GFF), CFVisual V2.1.5 software was used to visualize gene structure, conserved motifs, and intron–exon [51].

### 4.6. Analysis of Cis-Regulatory Elements of CCoAOMT Gene Promoter and Prediction of Secondary Structure of PgCCoAOMT Protein in P. grandiflorus

Based on the GFF annotation file of the *P. grandiflorus* genome and the full length of the *CCoAOMT* gene family, the upstream 2000 bp sequence of the promoter was extracted by TBtools software, and the cis-regulatory elements (CREs) of the promoter were predicted by PlantCARE, an online promoter analysis tool [52]. The HeatMap function of the TBtools software is used to visually analyze the number of cis-acting components. The secondary structure of the PgCCoAOMT protein was predicted by the SOPMA online website, and four conformations were obtained including alpha helix, extended strand, beta turn, and random coil [53].

### 4.7. Extraction, Quality Detection, and Real-Time Fluorescence Quantitative PCR (qRT-PCR) Verification of Total RNA from P. grandiflorus

According to the manufacturer’s instructions, Plant Total RNA Extraction Kit (Simgen Biotechnology Co., Ltd. Hangzhou, China) total RNA was extracted from *P. grandiflorus* samples treated with copper. The quality of total RNA was determined by 1% agarose gel electrophoresis. The cDNA was then obtained by the HiScript^®^ III RT SuperMix for qPCR (+gDNA wiper) reverse transcription kit, and the concentration of the obtained cDNA was determined by an ultraviolet–visible spectrophotometer. The primers used are designed by Primer3web version 4.1.0 (Appendix A) [54]. The expression of the *CCoAOMT* gene was detected by qRT-PCR using ChamQ Universal SYBR QPCR Master Mix (Nanjing Vazyme Biotech Co., Ltd. Nanjing, China). The PCR procedure is as follows: 95 °C, 30 s; 40 cycles, 95 °C, 10 s, 60 °C, 30 s. The melting curve of the sample is generated at 95 °C, 30 s, 65 °C, 30 s, and 95 °C, 30 s. PgGAPDG was used as the internal reference gene [55]. Relative gene expression was calculated using the 2^®−ΔΔCt^ method [56]. Each reaction was repeated three times and the results were expressed as the mean and variance of the three independent technical and biological replicates.

### 4.8. Prediction of CCoAOMT Protein Interaction Network in P. grandiflorus

The interaction network between the PgCCoAOMT protein and other proteins was analyzed using the STRING Version12.0 database. The minimum required interaction score was set to the highest confidence (0.900). At the same time, hide the disconnected nodes in the network, and output the TSV format file [57]. Import the file into Cytoscape3.10 software to beautify the protein interaction network diagram [58].

### 4.9. Determination of the Content of Lignin in P. grandiflorus Seedlings Under Copper Stress

For this experiment, the lignin content determination kit (Suzhou Grace Biotechnology Co., Ltd., Suzhou, China) was selected. This kit employs the acetylation method to cause the phenolic hydroxyl groups in lignin to undergo acetylation. It has a characteristic absorption peak at 280 nm, and the absorbance value at 280 nm is positively correlated with the lignin content. The lignin content of the seedings of *P. grandiflorus* samples was determined by the VICTOR^®^ Nivo^TM^ multi-mode microplate reader at the ultraviolet wavelength (280 nm). Take appropriate amounts of tissue samples from the copper stress treatment groups at 1 d, 3 d, 5 d, and 7 d as well as the control group and dry them at 45 °C for 3 days. The samples were ground in a mortar and passed through a 100-mesh sieve. An amount of 10 mg of the sieved powder tissue was weighed into a 1.5 mL centrifuge tube and 1.5 mL of 80% ethanol was added. The mixture was vortexed and shaken to mix evenly. The tube was placed in a water bath at 50 °C for 20 min and shaken every 3 min. After removal, the tube was cooled in running water to room temperature and centrifuged at 12,000 rpm for 10 min at room temperature. The supernatant was discarded and the precipitate was retained. An amount of 1 mL of 80% ethanol was added to the precipitate and shaken for 2 min to mix evenly. The same procedure was repeated. The precipitate was dried at 95 °C and the required reagents were added to the LABSELECT 96-well ultraviolet analysis plate in proportion according to the manufacturer’s instructions. Three replicate samples were measured for each treatment and the lignin content of the *P. grandiflorus* tissue samples was determined.

### 4.10. Statistical Analysis

Statistical analyses presented in this study included three technical replicates and biological replicates, and one-way ANOVA (and nonparametric or mixed) tests were used to determine *p*-values, with * indicating whether there was significance between the treated and blank groups (* *p* < 0.05, ** *p* < 0.01, *** *p* < 0.001, **** *p* < 0.0001).

## 5. Conclusions

In this study, the family of *P. grandiflorus CCoAOMT* genes was analyzed and five *PgCCoAOMT* genes were identified. Four genes were located across four chromosomes, with an additional gene found on Contig00572. Phylogenetic tree analysis showed that it was divided into two subgroups (Clade1a, Clade1b, Clade1c, Clade1d, and Clade2), all of which contained Motif1–4. Subcellular localization predicted that four PgCCoAOMT proteins were located in the cytoplasm and one in the chloroplasts. This study systematically revealed the phased regulatory patterns of the *PgCCoAOMT* gene in response to copper stress and its molecular association with the collapse of lignin synthesis through integrating the morphological characteristics of *P. grandiflorus* seedlings under copper stress, the results of qRT-PCR, and the trend of lignin content changes. This study first constructed the regulatory network of *CCoAOMT* genes in medicinal plants under heavy-metal stress, providing key insights for further understanding the lignin-mediated plant stress resistance.

## Figures and Tables

**Figure 1 ijms-26-04709-f001:**
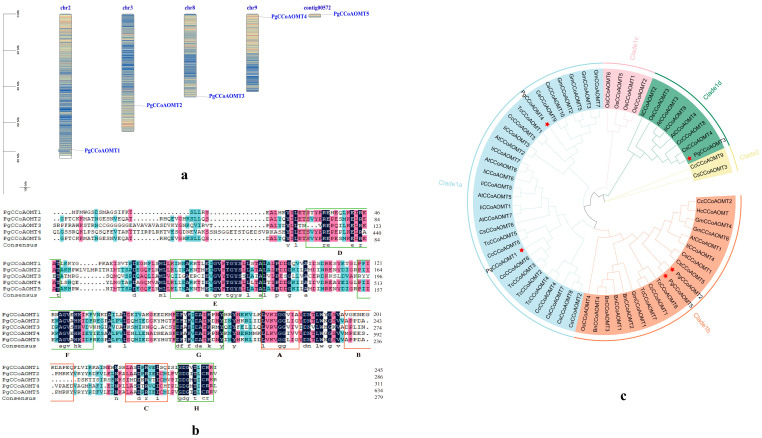
(**a**) Distribution of the *PgCCoAOMT* gene in *P. grandiflorus* genome. (**b**) Multiple sequence alignment results of five PgCCoAOMT proteins. The orange frame represents the common conserved sequences A, B, and C of plant methyltransferase; the green frame represents the tag sequences D, E, F, G, and H that are unique to the *CCoAOMT* gene family. The positions of highly conserved amino acids are marked with letters at the bottom. (**c**) Phylogenetic tree of members of the *CCoAOMT* gene family in 10 different plants. The red pentagram represents members of the *PgCCoAOMT* gene family.

**Figure 2 ijms-26-04709-f002:**
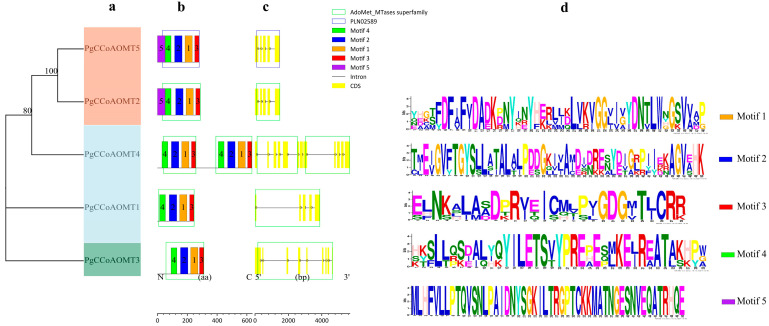
(**a**) Phylogenetic analysis, (**b**) protein motif, (**c**) gene structures, (**d**) display of the conserved motif sequence of the *CCoAOMT* gene family in *P. grandiflorus*.

**Figure 3 ijms-26-04709-f003:**
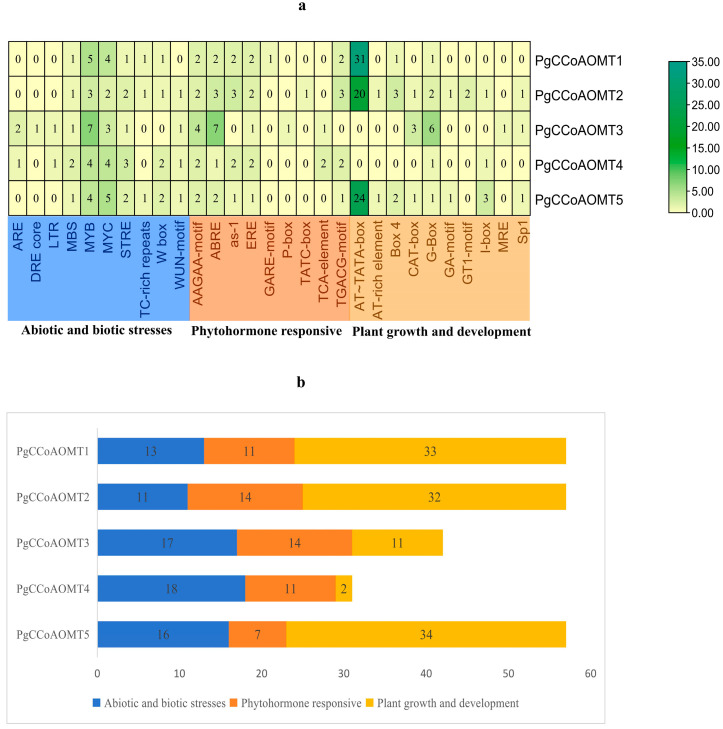
Analysis of cis-regulatory elements in the promoter region of the *CCoAOMT* gene of *P. grandiflorus.* (**a**) The horizontal coordinate represents different cis-regulatory elements, the vertical coordinate is the *CCoAOMT* gene, and the middle number represents the number of cis-regulatory elements. The larger the number, the higher the frequency of occurrence and the darker the green color. (**b**) It is a stacked bar chart, with blue representing elements related to environmental stress, orange representing elements related to hormonal response, and yellow representing elements related to growth and development.

**Figure 4 ijms-26-04709-f004:**
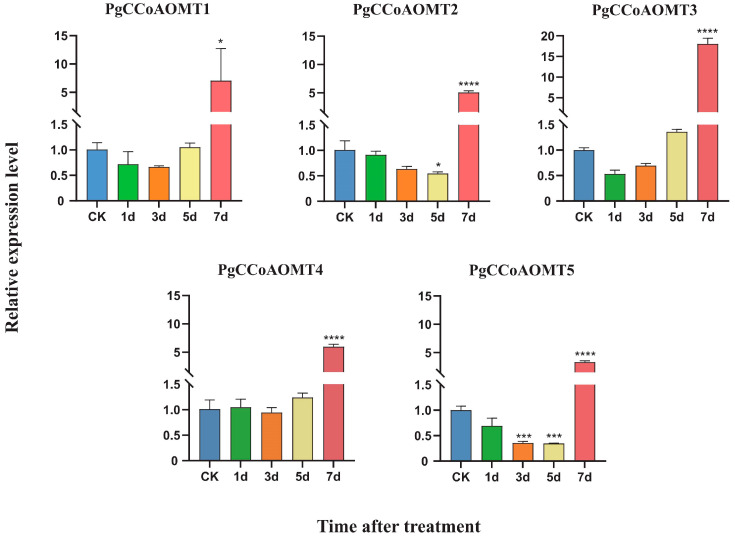
qRT-PCR analysis of five *PgCCoAOMT* genes under copper stress. One-way analysis of variance (ANOVA) was used to analyze the significance of copper stress treatment for 1 d, 3 d, 5 d, 7 d, and CK samples (* *p* < 0.05, *** *p* < 0.001, **** *p* < 0.0001).

**Figure 5 ijms-26-04709-f005:**
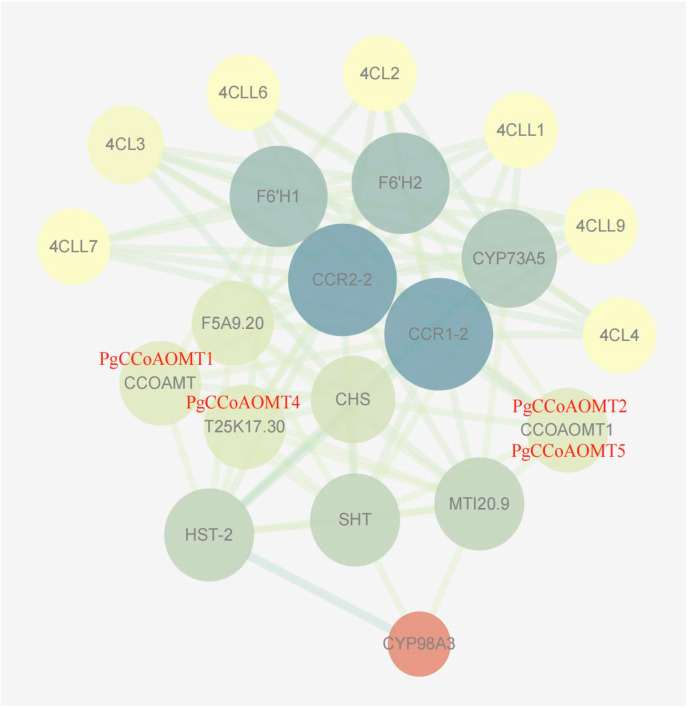
PgCCoAOMT protein functional interaction network. Note: The darker the line color, the stronger the interaction intensity; the lighter the line color, the weaker the interaction intensity.

**Figure 6 ijms-26-04709-f006:**
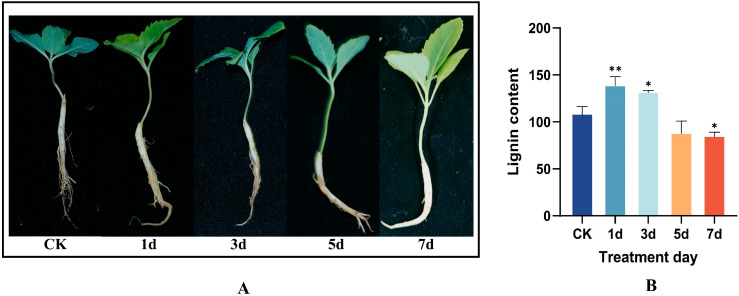
Characterization of *P. grandiflorus* seedlings under copper stress (1 d, 3 d, 5 d, 7 d were treated days) (**A**) and the changing trend of lignin content (**B**). (* *p* < 0.05, ** *p* < 0.01).

**Table 1 ijms-26-04709-t001:** The detailed information of genes and proteins of PgCCoAOMT.

Gene Name	Accession Number	Chromosome	Number of Amino Acids	Molecular Weight	Isoelectric Point	Subcellular Localization Predicted
PgCCoAOMT1	Pg_chr02_53430	chr2	246	27.66	5.56	Cytoplasmic
PgCCoAOMT2	Pg_chr03_29690	chr3	287	32.29	6.00	Cytoplasmic
PgCCoAOMT3	Pg_chr08_33400	chr8	311	34.02	9.15	Chloroplast
PgCCoAOMT4	Pg_chr09_01760	chr9	635	69.76	5.65	Cytoplasmic mitochondrial
PgCCoAOMT5	Pg_contig00572_00060	contig00572	280	31.46	5.83	Cytoplasmic

**Table 2 ijms-26-04709-t002:** Secondary structures of PgCCoAOMT proteins (%).

Protein Name	Alpha Helix	Extended Strand	Beta Turn	Random Coil
PgCCoAOMT1	38.62	17.89	0	43.50
PgCCoAOMT2	39.37	13.59	0	47.04
PgCCoAOMT3	38.26	14.79	0	46.95
PgCCoAOMT4	38.74	12.28	0	48.98
PgCCoAOMT5	33.57	14.29	0	52.14

## Data Availability

The original contributions presented in this study are included in the article/Appendix A. Further inquiries can be directed to the corresponding authors.

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
