# Peer review of "Genome-Wide Analysis of the Caffeoyl Coenzyme A-O-Methyltransferase (*CCoAOMT*) Gene Family in *Platycodon grandiflorus* (Jacq.) A. DC. and the Potential Regulatory Mechanism in Response to Copper Stress"

_ijms, 2025, doi:10.3390/ijms26104709_

Round 1

Reviewer 1 Report

Comments and Suggestions for Authors

Dear Authors,

Authors concentrated on caffeoyl coenzyme A-O-methyltransferase (CCoAOMT) gene family and potential regulatory mechanism in response to copper stress.

Authors indicated that five members of the PgCCoAOMT gene family was identified in P. grandiflorus. Moreover, promoter revealed that the PgCCoAOMT members contained a large number of cis-regulatory elements responsive to stress, and PgCCoAOMT2 and PgCCoAOMT5 were involved in the elements of lignin synthesis. Furthermore, Authors stated that the expression levels of all five PgCCoAOMT genes increased significantly at 7 days of treatment, suggesting that they might be involved in P. grandiflorus resistance.

Please, provide clear aim of the studies, because in current form we have it right away information what Author’s studies shown.

Panels a, b and c form Figure 1 should be enlarged, because in current form the reader lost important details – figure 2 the same situation.

The introduction gives the reader sufficient background to analyze Authors obtained results, but one issue should be deeply clarified: Why copper tolerance/stress is important in the context of  exactly P. grandifloras plants seedlings?

Why figure 6 are incorporated into discussion part?

What kind of “tissue samples” were used to lignin content detection – it should be added, it is important aspect when we analysed plants lignification level.

Moreover, what does it mean total RNA were isolated from P. grandifloras samples?

If I see properly part 2.7– figure 4 described RT-PCR , but not protein interactions (like figure 5) – Please, correct figures citation to correspond with described results;

How can the reviewer see the data about subcellular predicted localization? – Sorry, but I do not find it in manuscript as well as in supplements.

Moreover, if you add ANOVA test to variance analysis you should after use some statistical test to evaluate statistical significance - Please, provide information what kind of statistical test were used by Authors.

Sincerely

Author Response

Authors Response

Point-by-point responses to the reviewers’ comments:

We sincerely appreciate the thoughtful comments and recommendations from the editors and reviewers regarding our manuscript. These insights have been extremely helpful in revising and improving our paper. We have carefully considered the feedback and made the necessary revisions. We hope the updated manuscript aligns with the journal’s high standards. The main amendments are highlighted in blue font in the revised manuscript, and our responses to the reviewers’ comments are provided below:

Point-by-point responses to the reviewers’ comments:

Reviewer #1:

Authors concentrated on caffeoyl coenzyme A-O-methyltransferase (CCoAOMT) gene family and potential regulatory mechanism in response to copper stress.

Authors indicated that five members of the PgCCoAOMT gene family was identified in P. grandiflorus. Moreover, promoter revealed that the PgCCoAOMT members contained a large number of cis-regulatory elements responsive to stress, and PgCCoAOMT2 and PgCCoAOMT5 were involved in the elements of lignin synthesis. Furthermore, Authors stated that the expression levels of all five PgCCoAOMT genes increased significantly at 7 days of treatment, suggesting that they might be involved in P. grandiflorus resistance.

My comments are:

1、Please, provide clear aim of the studies, because in current form we have it right away information what Author’s studies shown.

Reply: We thank the reviewer for this valuable suggestion. We have supplemented the research objective under the Abstract in accordance with the reviewers' suggestions, hoping to facilitate the reading of both readers and reviewers. The modification is as follows:

This study reveals the expression level of PgCCoAOMT gene in the seedings of P. grandiflorus under copper stress, providing a theoretical basis for elucidating the mechanism of P. grandiflorus response to copper stress and for subsequent improvement of root resistance in P. grandiflorus.

2、Panels a, b and c form Figure 1 should be enlarged, because in current form the reader lost important details – figure 2 the same situation.

Reply: We thank the reviewer for this valuable suggestion. We will appropriately enlarge the modules a, b and c in Figures 1 and 2 by a certain scale. Hope this modification can facilitate the reading for the reviewers and readers.

3、The introduction gives the reader sufficient background to analyze Authors obtained results, but one issue should be deeply clarified: Why copper tolerance/stress is important in the context of exactly P. grandifloras plants seedlings?

Reply: We are extremely grateful to the reviewers for their detailed and professional feedback.During the seedling stage, plants exhibit vigorous growth and more obvious responses to stress, making it easier to observe morphological and physiological changes. Moreover, due to their physiological sensitivity and experimental operability, this study can efficiently reveal the dynamic regulatory network of the CCoAOMT gene family in lignin synthesis and stress resistance.

4、Why figure 6 are incorporated into discussion part?

Reply: We would like to express our sincere thanks to the reviewers for their valuable comments. According to the reviewers' suggestions, we have placed Figure 6 in Section 2.8 The trend of lignin content changes in the tissue samples of P. grandiflorus.We hope that our adjustments can bring convenience to the reviewers and readers.

5、What kind of “tissue samples” were used to lignin content detection – it should be added, it is important aspect when we analysed plants lignification level.

Reply: I would like to express my sincere appreciation to the reviewers for their comments. According to the reviewers' comments, we have supplemented the details of the copper stress treatment samples selected for the lignin content determination experiment. We dried the samples of the CK group, the copper stress group after 1 day, 3 days, 5 days, and 7 days respectively. The following changes have been made:

Take appropriate amounts of tissue samples from the copper stress treatment groups at 1d, 3d, 5d, 7d and the control group, dry them at 45°C for 3 days, grind them in a mortar, and filter through a 100-mesh sieve.

6、Moreover, what does it mean total RNA were isolated from P. grandifloras samples?

Reply: Thank you for the reviewer's patient and meticulous reading and evaluation. Extract RNA from the seedlings of P. grandiflorus. Then reverse transcribe it into cDNA. The cDNA can be used for further qPCR experiments to verify the expression analysis of PgCCoAOMT under different copper stress treatment days.

7、If I see properly part 2.7– figure 4 described RT-PCR , but not protein interactions (like figure 5) – Please, correct figures citation to correspond with described results;

Reply: We apologize to the reviewers for our oversight. We have made amendments to the figure caption in 2.7.

8、How can the reviewer see the data about subcellular predicted localization? – Sorry, but I do not find it in manuscript as well as in supplements.

Reply: Thank you very much for your careful review. In Section 4.3, we have described the website for predicting subcellular localization. Due to the malfunction of the confocal microscope in the laboratory, the subcellular localization experiment could not be carried out. Therefore, this study aimed to make up for this deficiency by means of bioinformatics. We apologize to the reviewers for this oversight.

Subcellular localization prediction of the PgCCoAOMT gene obtained by CELLO v.2.5 was performed.

9、Moreover, if you add ANOVA test to variance analysis you should after use some statistical test to evaluate statistical significance - Please, provide information what kind of statistical test were used by Authors.

Reply: We are very sorry that we did not clearly explain the meaning of one-way ANOVA in the materials and methods section of the paper. Your suggestions are of great significance to improve the science and readability of this article.

Reviewer 2 Report

Comments and Suggestions for Authors

The paper devoted to characterization of A-O-methyltransferase (CCoAOMT) Gene family in Platycodon and possible response to heavy metal (copper) stress.

Authors performed a good work, but the description and conclusions required significant re-evaluations.

Grammar and  logic need to be corrected.

Some details:

Lines 13 and 14: excatly the same words. Need to be improved.

Line 19_ „The analysis of the cis-regulatory element analysis“ ???

Line 22: „involved in the elements of lignin synthesis“ ??? Maybe „in lignin biosynthesis pathways“?

Line 24: „different expression paĴerns.“ Or level?

Line 26: „involved in P. grandiflorus resistance to copper stress.“ ? These genes is accompanied, but not directly invloved.

Line 48: : „toxic effects on plants. These effects include ion toxicity, oxidation stress, and osmotic stress,?  Effect – ion toxicity??

Line 48 – 51: more Details are required.

Line 58: „collected“ = described.

Lines 98 – 99_how did you identify it? Which methods?

Line 124: „were identified and identified“ ?? May be compared?

„Gene domains play a critical role in gene function” ¿? Domain can not play role in function itself.

Line 223: „ The protein secondary structure is the key to …the protein structure.“ ???

Line 234: some intriductiion how copper applied were required here, not only in M&M.

Lines 238 – 239: wrong conclusions. High expression level do not mean invlove. Moreover, as it is clear form figures, the expression level were down reguklated as respopnse. And increase after very long duration, at which so many genes change because of plant adaptaiion and global reprogramm.

Lines 257 – 260: this is acontradiction with your conclusion: at day 1-2 lignin is maximal, while gene expression is lower as control.

Line 281: “normal physiological activities.” ¿? What is it?

Line 290: „the PgCCoAOMT gene might participate in P. grandiflorus seedlings resisting copper stress by mediating lignin synthesis.“ ?? Higher lignin = lower mRNA, right? How you came to conclusion abiout invloved of gene in lignin under copper?

Line 366: “slow seedling cultivation” ¿? “7 days of slow seedling??

Line 481: „he defense against copper stress by mediating lignin synthesis,“ ???

Comments on the Quality of English Language

There are many repetions, not correct terms etc. see comments

Author Response

Authors Response

Point-by-point responses to the reviewers’ comments:

We sincerely appreciate the thoughtful comments and recommendations from the editors and reviewers regarding our manuscript. These insights have been extremely helpful in revising and improving our paper. We have carefully considered the feedback and made the necessary revisions. We hope the updated manuscript aligns with the journal’s high standards. The main amendments are highlighted in blue font in the revised manuscript, and our responses to the reviewers’ comments are provided below:

Point-by-point responses to the reviewers’ comments:

Reviewer #2:

The paper devoted to characterization of A-O-methyltransferase (CCoAOMT) Gene family in Platycodon and possible response to heavy metal (copper) stress.

Authors performed a good work, but the description and conclusions required significant re-evaluations.

Grammar and logic need to be corrected.

My comments are:

Some details:

1、Lines 13 and 14: excatly the same words. Need to be improved.

Reply: Thank you for the reviewers' comments. We have consolidated the sentences in line 13 and line 14 that are identical. We apologize for our oversight. The changes we have made are as follows:

In this study, the CCoAOMT gene family members of Platycodon grandiflorus were identified by bioinformatics methods, and their basic characteristics and potential functions were analyzed.

2、Line 19_ „The analysis of the cis-regulatory element analysis“ ???

Reply: We would like to express our gratitude to the reviewers for their patient review. We have corrected the issues pointed out by the reviewers. At the same time, we have checked and revised the possible problems in the article. Once again, we apologize to you for our oversight. The changes we have made are as follows:

The cis-regulatory element analysis of the promoter revealed that the PgCCoAOMT members contained a large number of cis-regulatory elements responsive to stress, and PgCCoAOMT2 and PgCCoAOMT5 were involved in the elements of lignin synthesis.

3、Line 22: „involved in the elements of lignin synthesis“ ??? Maybe „in lignin biosynthesis pathways“?

Reply: We would like to express our gratitude to the reviewer for his/her meticulous guidance. We apologize for the inappropriate description in line 22 of the abstract. At the same time, we re-analyzed the lignin content and qPCR results in the results section. By reading multiple papers, we re-evaluated the descriptions and conclusions presented in the article.

4、Line 24: „different expression paĴerns.“ Or level?

Reply: Thank you for the reviewers' suggestions. We have corrected the possible inappropriate expressions in the article and once again express our gratitude for the reviewers' meticulous review. Hereby the changes are as follows:

The qRT-PCR results showed that, within 5 days of copper stress treatment, except for the PgCCoAOMT4 gene, the other genes exhibited different expression levels. Furthermore, the expression levels of all five PgCCoAOMT genes increased significantly at 7 days of treatment.

5、Line 26: „involved in P. grandiflorus resistance to copper stress.“ ? These genes is accompanied, but not directly invloved.

Reply: Thank you for the reviewers' comments. We have revised the content of the article. We found that the expression of some CCoAOMT genes might be inhibited by certain transcription factors under copper stress. Therefore, it is particularly important to explore the molecular mechanism under copper stress in Platycodon grandiflorum. The inappropriate description here has been deleted.

6、Line 48: : „toxic effects on plants. These effects include ion toxicity, oxidation stress, and osmotic stress,?  Effect – ion toxicity??

Reply: We are extremely grateful for the patience and careful review of the reviewer. We sincerely apologize for the problems existing in the writing. We promptly made revisions according to the reviewer's suggestions and deleted some inappropriate expressions. The questions raised by you have provided guiding opinions for the revision of this article.

7、Line 48 – 51: more Details are required.

Reply: Thank you for the reviewers' comments. We have added three relevant references on the effects of copper deficiency and excess on plants in lines 48-51 of the article. Once again, we are grateful for the reviewers' comments, which have made the logic of the introduction of the article clearer. The supplementary references are as follows:

[8]Zhang L, Wang Y, Yang D, et al. Platycodon grandiflorus - an ethnopharmacological, phytochemical and pharmacological review. J Ethnopharmacol. 2015; 164: 147-161.

[9]Zhang Y, Sun M, He Y, et al. Polysaccharides from Platycodon grandiflorum: A review of their extraction, structures, modifications, and bioactivities. Int J Biol Macromol. 2024; 271: 132617.

[10]Luo Z, Xu W, Yuan T, et al. Platycodon grandiflorus root extract activates hepatic PI3K/PIP3/Akt insulin signaling by enriching gut Akkermansia muciniphila in high fat diet fed mice. Phytomedicine. 2023; 109: 154595.

8、Line 58: „collected“ = described.

Reply: We are very grateful for the reviewers' comments. We have changed "collected" to "described" in the text. We sincerely apologize for the inappropriate wording at line 58. We also thank the reviewers for their patient review. We hope that our changes can bring convenience to readers' reading.

9、Lines 98 – 99_how did you identify it? Which methods?

Reply: Thank you for the questions raised by the reviewers. The method for identifying members of the CCoAOMT gene family of Aristolochia is described in Section 4.3 of the Materials and Methods section of the text. Using A. thaliana protein sequences as bait sequences, snake bed protein sequences were searched by bidirectional blast comparison using TBtools software, and preliminary candidate sequences of members of the PgCCoAOMT gene family were obtained(Chen, et al. 2023). The sequences obtained were integrated by two methods to identify the final members of the P. grandiflorus CCoAOMT gene family. On the one hand, from the conservative InterPro database (https://www.ebi.ac.uk/interpro/search/sequence/) to download the CCoAOMT gene structure domain hidden Markov model (PF01596). Then the candidate protein sequences of the CCoAOMT gene family were screened by HMMER3.1 software (e-value <1×10−20)(Blum, et al. 2025; Finn, et al. 2011). On the other hand, the NCBI CDD online site is used to manually eliminate redundant sequences(Wang, et al. 2023). The sequences obtained by the above two methods were integrated to obtain the final members of the CCoAOMT gene family of P. grandiflorus.

10、Line 124: „were identified and identified“ ?? May be compared?

Reply: Thank you for the reviewers' questions. We have made a mistake in our expression which deviated from the content of the article. We are very sorry for this. We have made the following changes:

In this study, the five identified PgCCoAOMT members were used to construct a phylogenetic tree together with 58 CCoAOMT members from A. thaliana、B. nivea、C. capsularis、C. sinensis、G. max、H. cannabinus、I. indigotica、O. sativa、T. cacao.

11、„Gene domains play a critical role in gene function” ¿? Domain can not play role in function itself.

Reply: Thank you for your patient review. We sincerely apologize to you for any mistakes in our writing. We have revised the article to correct the possible grammatical errors and inappropriate expressions. The revisions are as follows:

Analysis the functions of gene domains is fundamental to comprehending the regulatory mechanisms of life.

12、Line 223: „ The protein secondary structure is the key to …the protein structure.“ ???

Reply: Thank you for the reviewers' corrections. We have revised the inappropriate expression in this sentence and hope it can bring convenience for readers' reading. The changes are as follows:

Proteins secondary structure is the fundamental basis for maintaining the structural stability and realizing the biological functions of proteins.

13、Line 234: some intriductiion how copper applied were required here, not only in M&M.

Reply: Thank you for the reviewers' comments. We have added some details at the original line 234 of the manuscript to enrich this part. This addition is not only reflected in the Materials and Methods section but also elsewhere. The details are as follows:

The expression of the PgCCoAOMT gene under copper stress at 1d, 3d, 5d and 7d was verified by qRT-PCR analysis compared with the CK group (Fig. 4).

14、Lines 238 – 239: wrong conclusions. High expression level do not mean invlove. Moreover, as it is clear form figures, the expression level were down reguklated as respopnse. And increase after very long duration, at which so many genes change because of plant adaptaiion and global reprogramm.

Reply: Thank you for the reviewers' comments. Your suggestions provided guiding opinions for the revision of the article. We have rewritten the results in Section 2.6. The changes are as follows:

The expression of the PgCCoAOMT gene under copper stress was verified by qRT-PCR analysis (Fig. 4). The results showed that all genes except PgCCoAOMT4 showed different expression patterns after 5 days of copper stress treatment. The expression of PgCCoAOMT1 and PgCCoAOMT3 shows a trend of inhibition followed by activation, suggesting that they are involved in the staged repair process. The early down-regulation of these genes under copper stress treatment may be for resource conservation in metabolism, while the later up-regulation is to cope with the cell wall damage caused by copper ion permeation. Both PgCCoAOMT2 and PgCCoAOMT5 showed down-regulated expression. It is speculated that the promoter regions of these two genes may contain stress-sensitive elements. In the early stage of copper stress, they are inhibited by transcription factors and prioritize the allocation of resources to the antioxidant pathway. The expression levels of five PgCCoAOMT genes increased significantly after 7 days of treatment.It is speculated that the metabolic collapse caused by root damage resulting from long-term stress leads to the failure of gene expression upregulation.

15、Lines 257 – 260: this is acontradiction with your conclusion: at day 1-2 lignin is maximal, while gene expression is lower as control.

Reply: We are extremely grateful for the reviewers' comments. We are deeply appreciative of the guiding suggestions put forward by the reviewers. By reading multiple literatures and combining the morphological characteristics of the young roots of Platycodon grandiflorum under copper stress, the results of qRT-PCR and the trend of lignin content change in this experiment, we have rewritten the relevant content. We hope that the revised content can meet the reading needs of readers.

16、Line 281: “normal physiological activities.” ¿? What is it?

Reply: We are extremely grateful for the patience and careful review of the reviewer. We have corrected the mistakes in the expression. The modifications are as follows: 

It was speculated that plants may accumulate lignin to strengthen cell walls and restrict ion permeability.

17、Line 290: „the PgCCoAOMT gene might participate in P. grandiflorus seedlings resisting copper stress by mediating lignin synthesis.“ ?? Higher lignin = lower mRNA, right? How you came to conclusion abiout invloved of gene in lignin under copper?

Reply: Thank you for the reviewers' questions. We incorporated your suggestions into the discussion of the article by adding 3.3 The potential role of PgCCoAOMT gene under copper stress. Meanwhile, we removed the original 3.3. We believe that this lagging gene expression and the metabolic phenotype of rapid lignin accumulation imply the existence of a multi-level regulatory network in the seedings of P. grandiflorus under copper stress. The changes are as follows:

The qRT-PCR results indicated (Figure 4) that within 5 days of copper stress treatment, members of the PgCCoAOMT gene family showed different expression patterns. The expression of PgCCoAOMT1 and PgCCoAOMT3 showed a trend of inhibition followed by activation, while PgCCoAOMT2 and PgCCoAOMT5 showed continuous down-regulation, and the expression of PgCCoAOMT4 showed no significant change. Notably, at the long-term stress (7 days) stage, the expression levels of all PgCCoAOMT genes were significantly up-regulated. However, the lignin content showed a trend of increasing first and then decreasing. This lag in gene expression and the rapid accumulation of lignin content jointly suggested that there was a multi-level regulatory network in the seedings of P. grandiflorus under copper stress. Therefore, we speculate that under the 1-3 days of stress treatment, plants initiated a rapid response, showing rapid accumulation of lignin content. This might be dependent on the early accumulation of precursor substances or the spontaneous activation of other biosynthetic pathways of lignin. The brief inhibition of PgCCoAOMT1 and PgCCoAOMT3 might be a resource optimization strategy, allocating metabolic fluxes preferentially to the antioxidant system to resist ROS bursts. At the 3-5 days of stress treatment, metabolic imbalance occurred in the seedings of P. grandiflorus. PgCCoAOMT2 and PgCCoAOMT5 showed continuous inhibition, which might lead to the blockage of the key step of lignin methylation and affect the subsequent precursor supply, thereby causing the gradual decrease in lignin content. Finally, under the 5-7 days of copper stress treatment, the metabolic network in the plant body showed a systemic collapse. Although the overall expression of PgCCoAOMT genes was significantly up-regulated, it was unable to reverse the lignin synthesis and accumulation in the plant body. Specifically, the lignin content was significantly lower than the CK group on day 7, and leaf scorch and root rot diseases occurred.

18、Line 366: “slow seedling cultivation” ¿? “7 days of slow seedling??

Reply: We are extremely grateful for the reviewers' comments. We have made corrections to two errors based on their suggestions, hoping that these changes will facilitate the reading of the reviewers and readers. We apologize again for our oversight. The changes are as follows:

reducing transplant shock;

acclimatization of transplanted seedlings.

19、Line 481: „he defense against copper stress by mediating lignin synthesis,“ ???

Reply: We are grateful for the reviewers' comments. We have revised the inappropriate expression in line 481 of the conclusion of the article based on the revised content. Thank you again for your valuable comments!

This study systematically revealed the phased regulatory patterns of the PgCCoAOMT gene in response to copper stress and its molecular association with the collapse of lignin synthesis through integrating the morphological characteristics of P. grandiflorus seedlings under copper stress, the results of qRT-PCR, and the trend of lignin content changes. This study first constructed the regulatory network of CCoAOMT genes in medicinal plants under heavy metal stress, providing key insights for further understanding the lignin-mediated plant stress resistance.

Reviewer 3 Report

Comments and Suggestions for Authors
  1. The manuscript titled as “Genome-wide Analysis of the Caffeoyl Coenzyme A-O-methyltransferase (CCoAOMT) Gene family in Platycodo grandiflorus (Jacq.) A. DC. and the Potential Regulatory Mechanism in Response to Copper Stress” is very well written. The study focused on understanding the functions of CCoAOMT gene in P. grandifloras and its role in enhancing plant resistance to copper stress. The authors identified 5 different genes related to CCoAOMT distributed over 4 chromosomes and Contig00572. In qRT-PCR experiments, the authors found out that the exoression of CCoAOMT gene decreases at early time points of copper stress but then increases at day 7 suggesting their potential role in providing defense against copper stress. The study also reported the change in levels of lignin production over time in response to copper stress. Overall, the experimental design is sound, and the results are clearly presented. The findings contribute to our understanding of functions and defense role of CCoAOMT gene in response to copper stress.
  2. In the abstract, please explicitly mention copper stress as among the main problems being addressed in this paper.
  3. There is a rationale missing in the introduction about why CCoAOMT is being chosen for addressing the copper problem. Is there any relevant literature citing the role of CCoaAOMT in providing resistance to copper stress? If so, please cite the relevant studies. Also mention and expand on how lignin is important during copper stress to set the stage for the readers.
  4. Line 60: Change Platycodon grandifloras to grandifloras
  5. Line 78: Replace kind by “type”
  6. Line 83: Please change “phenylpropane pathway” to “phenylpropanoid pathway”.
  7. Line 256-261: Please cite the corresponding figure related to the text mentioned in the Results section to improve data interpretation
  8. Line 365: Please mention the ingredients in Hoagland nutrient solution.
  9. Line 369: Please mention on what basis 200mM CuSO4.5H2O concentration is considered for the experiment. It would have been informative and good comparison if normal concentration of Copper which is non stressful to the plant was used in the experiment. Without a normal copper control, it could be hard to distinguish between responses that happen under normal conditions versus those that are truly induced by excess copper (toxicity responses). Alternatively, if Hoagland solution contains copper, it would be good to mention for clarity.
  10. Line 399-401: Please italicize all the plant species.
  11. Line 462: Please specify which reagent was used. It is important to provide a clear and detailed methodology to ensure reproducibility of the experiment for the reader. Also mention, which standard was used for lignin quantification.

Author Response

Authors Response

Point-by-point responses to the reviewers’ comments:

We sincerely appreciate the thoughtful comments and recommendations from the editors and reviewers regarding our manuscript. These insights have been extremely helpful in revising and improving our paper. We have carefully considered the feedback and made the necessary revisions. We hope the updated manuscript aligns with the journal’s high standards. The main amendments are highlighted in blue font in the revised manuscript, and our responses to the reviewers’ comments are provided below:

Point-by-point responses to the reviewers’ comments:

Reviewer #3:

The manuscript titled as “Genome-wide Analysis of the Caffeoyl Coenzyme A-O-methyltransferase (CCoAOMT) Gene family in Platycodo grandiflorus (Jacq.) A. DC. and the Potential Regulatory Mechanism in Response to Copper Stress” is very well written. The study focused on understanding the functions of CCoAOMT gene in P. grandifloras and its role in enhancing plant resistance to copper stress. The authors identified 5 different genes related to CCoAOMT distributed over 4 chromosomes and Contig00572. In qRT-PCR experiments, the authors found out that the exoression of CCoAOMT gene decreases at early time points of copper stress but then increases at day 7 suggesting their potential role in providing defense against copper stress. The study also reported the change in levels of lignin production over time in response to copper stress. Overall, the experimental design is sound, and the results are clearly presented. The findings contribute to our understanding of functions and defense role of CCoAOMT gene in response to copper stress.

My comments are:

  • In the abstract, please explicitly mention copper stress as among the main problems being addressed in this paper.

Reply: We are grateful to the reviewers for their insightful comments on the abstract of the paper. Based on their suggestions, we have supplemented the issues related to copper stress in the abstract of the article. We hope that our revisions can facilitate readers' reading of this paper. The changes are as follows:

In recent years, copper pollution has gradually become one of the major problems of soil environmental pollution. Lignin plays an important role in plant resistance to biotic and abiotic stresses. CCoAOMT is a key enzyme in the lignin biosynthesis process.

2、There is a rationale missing in the introduction about why CCoAOMT is being chosen for addressing the copper problem. Is there any relevant literature citing the role of CCoaAOMT in providing resistance to copper stress? If so, please cite the relevant studies. Also mention and expand on how lignin is important during copper stress to set the stage for the readers.

Reply: We are grateful to the reviewers for their in-depth review and constructive suggestions on the paper, which are of vital importance for enhancing the research quality of this paper. In the third paragraph of the introduction, we have supplemented the information about the role of the CCoAOMT gene in resisting stress. The changes are as follows:

The CCoAOMT1 gene in A. thaliana regulates the accumulation of H2O2 and the responses of ABA and ROD signaling pathways to drought stress by modulating the drought resistance(Chun, et al. 2021). The ZmCCoAOMT2 in maize can regulate the content of H-lignin and programmed cell death (PCD), and plays an important role in resisting some diseases(Yang, et al. 2017).

3、Line 60: Change Platycodon grandifloras to grandifloras

Reply: We sincerely appreciate the key issues pointed out by the reviewers, which have significantly enhanced the scientificity and rigor of the research. We have changed the original 60 lines of "Platycodon grandifloras" to "P. grandifloras". At the same time, we have carefully checked the plants mentioned in the text to ensure accuracy.

4、Line 78: Replace kind by “type”

Reply: Thank you for your patient review. Your meticulous review has made the presentation of this article more rigorous and convenient for readers to understand.

5、Line 83: Please change “phenylpropane pathway” to “phenylpropanoid pathway”.

Reply: We are extremely grateful for the reviewers' comments. Based on their suggestions, we have changed "phenylpropane pathway" to "phenylpropanoid pathway". Once again, we would like to express our gratitude to the reviewers for their patient review.

6、Line 256-261: Please cite the corresponding figure related to the text mentioned in the Results section to improve data interpretation

Reply: We sincerely thank the reviewers for their valuable comments and suggestions. According to the reviewers' opinions, we have adjusted the position of Figure 6. We have placed Figure 6 under Section 2.8 so that readers can read this paper intuitively.

7、Line 365: Please mention the ingredients in Hoagland nutrient solution.

Reply: Thank you for the reviewers' comments. The 1/2 Hoagland nutrient solution used in this study was purchased from Coolaber. The components are as follows:

8、Line 369: Please mention on what basis 200mM CuSO4.5H2O concentration is considered for the experiment. It would have been informative and good comparison if normal concentration of Copper which is non stressful to the plant was used in the experiment. Without a normal copper control, it could be hard to distinguish between responses that happen under normal conditions versus those that are truly induced by excess copper (toxicity responses). Alternatively, if Hoagland solution contains copper, it would be good to mention for clarity.

Reply: We sincerely thank the reviewers for carefully reviewing the manuscript and providing valuable professional opinions. This study explored the optimal concentration and treatment time by integrating a large number of literatures and conducting preliminary pre-experiments. We discovered that 200mM CuSO4.5H2O can effectively trigger a series of physiological, biochemical and molecular responses in plants. Therefore, 200mM CuSO4.5H2O was selected in this study to simulate the copper stress problem that may be encountered in the production of P. grandifloras.

Meanwhile, we noticed that 1/2 Hoagland nutrient solution contains copper ions, but the author believes that this concentration is within the normal growth and development range for plants. Thus, it is expected that copper stress will not have an impact on this study. Once again, we would like to thank the reviewers for their patient review.

  • Line 399-401: Please italicize all the plant species.

Reply: Thanks very much for the reviewer's suggestion. According to the reviewers' comments, we have changed all the plant species in lines 399-401 to italic. Once again, we apologize for our oversight.

10、Line 462: Please specify which reagent was used. It is important to provide a clear and detailed methodology to ensure reproducibility of the experiment for the reader. Also mention, which standard was used for lignin quantification.

Reply: We are extremely grateful to the reviewers for their patient and meticulous review. The reagent kit used in this study is the Lignin Content Determination Reagent Kit (Suzhou Grise Biotechnology Co., Ltd., Suzhou, China). This reagent kit uses the acetylation method to cause the phenolic hydroxyl groups in lignin to undergo acetylation reactions. It has a characteristic absorption peak at 280 nm, and the absorbance value at 280 nm is positively correlated with the lignin content. Once again, we thank the reviewers for their guiding opinions, which have brought convenience for readers to read this article.

Round 2

Reviewer 2 Report

Comments and Suggestions for Authors

Thank you! The text is better,  conclusions became more clear, but you need to come through text once more time for polishing.

For example: Line 499: "4 genes were distributed on 4 chromosomes and 1 gene was distributed on Contig00572" ??

Line 386: " growth status and consistent growth" ??

Author Response

Authors Response

Thank you for your letter and for the reviewers comments concerning our manuscript entitled 'Genome-wide Analysis of the Caffeoyl Coenzyme A-O-methyltransferase (CCoAOMT) Gene family in Platycodon grandiflorus (Jacq.) A. DC. and the Potential Regulatory Mechanism in Response to Copper Stress'(ID: ijms-3516938). Those comments are all valuable and very helpful for revising and improving our paper, as well as the important guiding significance to our researches. We have studied comments carefully and have made correction which we hope meet with approval.

Point-by-point responses to the reviewers’ comments:

Reviewer #2:

Thank you! The text is better,  conclusions became more clear, but you need to come through text once more time for polishing.

My comments are:

1、For example: Line 499: "4 genes were distributed on 4 chromosomes and 1 gene was distributed on Contig00572" ??

Reply: We sincerely thank the reviewers for their careful reading and valuable suggestions. The text has been revised and improved item. Regarding the incorrectly expressed sentences in line 499, it has been modified as follows:

Four genes were located across four chromosomes, with an additional gene found on Contig00572.

2、Line 386: " growth status and consistent growth" ??

Reply: We sincerely appreciate the reviewers' insightful comments. All suggestions have been systematically addressed in the revision. Once again, apologize to you for our negligence. Revised as follows:

Seedlings with good growth status were selected and transplanted into incubators containing 1/2 Hoagland nutrient solution for reducing transplant shock, and 1 plant per hole was planted.